# Extracellular Vesicles, Inflammation, and Cardiovascular Disease

**DOI:** 10.3390/cells11142229

**Published:** 2022-07-18

**Authors:** Akbarshakh Akhmerov, Tanyalak Parimon

**Affiliations:** 1Smidt Heart Institute, Cedars-Sinai Medical Center, Los Angeles, CA 90048, USA; 2Women’s Guild Lung Institute, Pulmonary and Critical Care Division, Department of Medicine, Cedars-Sinai Medical Center, Los Angeles, CA 90048, USA; tanyalak.parimon@cshs.org

**Keywords:** extracellular vesicles, exosomes, microvesicles, inflammation, innate immunity, adaptive immunity, cardiovascular disease, cardiac disease, heart disease, myocardium

## Abstract

Cardiovascular disease is a leading cause of death worldwide. The underlying mechanisms of most cardiovascular disorders involve innate and adaptive immune responses, and extracellular vesicles are implicated in both. In this review, we describe the mechanistic role of extracellular vesicles at the intersection of inflammatory processes and cardiovascular disease. Our discussion focuses on atherosclerosis, myocardial ischemia and ischemic heart disease, heart failure, aortic aneurysms, and valvular pathology.

## 1. Introduction

Cardiovascular disease is a leading cause of morbidity and mortality, accounting for one-third of global deaths in 2019 [1]. Collectively, atherosclerosis, ischemic heart disease, heart failure, aortopathy, and valvular pathology make up most of these deaths (Figure 1). Inflammation plays a significant role in the pathogenesis of these cardiovascular conditions [2,3,4], and anti-inflammatory therapies have demonstrated benefits in several clinical trials [5,6]. The underlying mechanisms of inflammation in cardiovascular disease are multifaceted, but an emerging body of evidence implicates extracellular vesicles (EVs) as key mediators.

EVs are nano-sized, membrane-bound vesicles released by nearly all cells. EVs are extremely heterogeneous but can be broadly classified based on their size and biogenesis [7]. Exosomes are typically 50–150 nm in size, derived through an endosomal pathway, and released upon exocytosis of multivesicular bodies [8,9,10]. Microvesicles (also referred to as ectosomes or microparticles) are typically 100–1000 nm in size and are assembled at and released from the plasma membrane [10,11]. Similarly, apoptotic bodies are >100 nm in size but are formed during apoptosis. EVs are crucial in cell-to-cell communication and mediate their effects through bioactive cargo, including proteins, RNA, and lipids. The composition of EVs, however, may change depending on the physiologic state of the parent cell.

The role of EVs in inflammatory pathways and their involvement in cardiovascular disease have been independently discussed in other reviews [12,13,14,15]. Here, we review findings that implicate EVs at the intersection of inflammation and cardiovascular disease. As such, we highlight the inextricable interplay between EVs, inflammation, and cardiovascular disease (Graphical Abstract).

## 2. The Role of EVs and Inflammation in Specific Cardiovascular Disorders

### 2.1. The Role of EVs and Inflammation in Atherosclerosis

Atherosclerosis is the accumulation of fibrofatty material within an arterial wall, which can narrow or occlude the vessel lumen, impede blood flow, and ultimately result in myocardial infarction, stroke, or peripheral arterial disease [16]. The clinical sequelae of atherosclerosis (e.g., coronary artery disease, stroke) are the leading cause of death worldwide [1]. Although the pathogenesis of atherosclerosis is multifactorial, inflammation plays a central role [17]. Indeed, atherosclerosis is often characterized as a chronic inflammatory disorder.

Atherosclerosis is initiated by endothelial activation and dysfunction, with subsequent accumulation of lipid content (low-density lipoprotein, LDL) in the subendothelial layer of the artery. The lipid material undergoes oxidation and further modifications that propagate inflammatory responses driven by innate and adaptive immunities [16,17]. EVs have been isolated from intimal lesions and are a major determinant of atherosclerotic plaque progression [18]. Plaque analysis in humans showed that most EVs originate from leukocytes, with macrophages accounting for the majority (29%), followed by lymphocytes (15%) and granulocytes (8%) [19]. Smooth muscle cells, platelets, adipocytes, and endothelial cells, however, are equally important sources of EVs [18]. Regardless of their origin, EVs interact with immune cell mediators at all stages of disease progression (Figure 2) [20].

During the initial stages of atherogenesis, plaque EVs transfer the intercellular adhesion molecule 1 (ICAM-1) to endothelial cells, thereby promoting monocyte adhesion and recruitment [21]. Similarly, monocyte-derived EVs induce the expression of ICAM-1 and chemoattractant C–C motif chemokine ligand 2 (CCL2) by activating the nuclear factor-κB (NF-κB) pathway within endothelial cells [22]. Neutrophil-derived EVs also facilitate recruitment, by enhancing transendothelial migration of monocytes through upregulation of CCL2, ICAM-1, and the vascular cell adhesion molecule 1 (VCAM-1). The proposed mechanism involves activation of the NF-κB pathway by EV-delivered miR-155 [23]. Among non-leukocytes, activated platelets release EVs that facilitate monocyte recruitment by transferring the chemokine RANTES (Regulated upon Activation, Normal T Cell Expressed and Presumably Secreted) to the endothelium, and by transferring a platelet-adhesion molecule GPIbα directly to monocytes [24,25]. In addition, platelet-derived EVs promote neutrophil adhesion to endothelial cells through P-selectins [26]. In the setting of inflammation, adipocyte-derived EVs upregulate the expression of VCAM-1 on endothelial cells and enhance leukocyte attachment [27]. Finally, EVs released from endothelial cells pretreated with reactive oxygen species stimulate endothelial cells to bind monocytes [28]. Taken together, EVs released from activated leukocytes, platelets, endothelial cells, and adipocytes promote inflammatory infiltration. It is important to note, however, that most of these studies were performed in vitro, and further investigation is needed to firmly establish these mechanistic links.

Beyond the recruitment of leukocytes, EVs play an important role in plaque maturation. The polarization of macrophages and the progression of macrophages to foam cells are important examples. Following LDL deposition in the subendothelial region, reactive oxygen species convert LDL to oxidized LDL, which, in turn, leads to monocyte-to-macrophage differentiation [20,29]. Further uptake of oxidized LDL by macrophages via scavenger receptors leads to the formation of lipid-containing foam cells. Platelet-derived EVs promote phagocytosis of oxidized LDL by macrophages, thereby enhancing foam cell formation [30]. Adipose-derived EVs also enhance foam cell formation, by inhibiting cholesterol efflux from macrophages. Furthermore, these adipose-derived EVs polarize macrophages toward the classic pro-inflammatory M1 phenotype [31]. Similarly, endothelial EVs have the capacity to polarize macrophages, either toward the M1 phenotype or the anti-inflammatory M2 phenotype, depending on environmental stimuli [20]. Endothelial cells exposed to oxidized LDL secrete EVs containing metastasis-associated lung adenocarcinoma transcript 1 (MALAT1), which promotes M2 polarization [32]. Other studies, however, demonstrated that endothelial cells exposed to oxidized LDL secrete EVs containing miR-155, which shift the phenotype toward the M1 phenotype [33]. Given the potentially competing mechanisms, the overall effect of endothelium-derived EVs on macrophage polarization requires further investigation. 

The other important protein involved in LDL metabolism is heparan sulfate proteoglycan (HSPG), which offers a protective role in atherosclerosis. It is a key regulator of LDL retention in endothelial cells and immune cells (macrophages and neutrophils), given its constitutive expression in those cells [34]. HSPG halts the atherosclerotic process by binding to a proliferation-inducing ligand (APRIL), and serum levels of APRIL are associated with cardiovascular mortality in atherosclerotic patients [35]. In addition, HPSG facilitates endocytosis and uptake of LDLRQ722*-containing EVs, improving LDL clearance and preventing atherosclerotic plaque formation [36]. Overall, the presence of HPSG appears beneficial in atherosclerotic blood vessels. On the hand, proteoglycans in the extracellular matrix of the vascular wall may contribute to the pathogenesis of atherosclerosis, but it is unclear whether this process is mediated by EVs [37].

Another crucial component of atherosclerotic plaque progression is calcification, and EVs mechanistically contribute to this process [38]. Analysis of murine and human plaques showed that EVs derived from pro-inflammatory macrophages promote intimal microcalcification [39,40]. Furthermore, enrichment of these EVs with the calcium-binding protein S100A9 facilitates the development of calcification. Smooth-muscle-cell-derived EVs are equally important in promoting microcalcification. The reader is referred elsewhere for a thorough review of these mechanisms [20]. In conclusion, the interplay between EVs derived from leukocytes, platelets, endothelial cells, adipocytes, and smooth muscle cells is central to the pathogenesis of atherosclerosis, and mechanistic insights into this interplay have important therapeutic implications. When atherosclerotic disease progresses, the sequelae include myocardial infarction (MI) and ischemic heart disease.

### 2.2. The Role of EVs and Inflammation in Myocardial Infarction and Ischemic Heart Disease

Ischemic heart disease is a leading cause of mortality worldwide, with an estimated prevalence of 197 million cases in 2019 [41]. Inflammation and fibrosis play crucial, interconnected roles in the pathophysiology of MI and ischemic heart disease. EVs derived from cardiomyocytes, endothelial cells, and immune cells influence inflammatory and fibrotic responses and can therefore serve as potential diagnostic, prognostic, and therapeutic agents.

Following an acute MI, rapid humoral and cell-mediated responses ensue [42]. In preclinical models, infiltrating monocytes that subsequently differentiate into macrophages are the major cell mediators [43]. The release of EVs within the myocardium appears to correspond to the inflammatory influx, with a significant increase in EVs at 15–24 h following MI [44]. These EVs originate locally from cardiomyocytes and endothelial cells and promote the release of chemokines and inflammatory cytokines from infiltrating monocytes [44]. Similarly, cardiomyocyte-derived EVs modulate the inflammatory phenotype of macrophages, depending on the state of donor cells (i.e., ischemic versus non-ischemic) [45]. The crosstalk between cardiomyocytes also extends beyond the local milieu of the heart (Figure 3). Following acute MI, myocardial micro-RNAs (miRNAs) are selectively shuttled by EVs to peripheral organs. In mice, miR-1, miR-208, and miR-499 are transported by EVs to the bone marrow, where they downregulate the expression of CXC chemokine receptor 4 (CXCR4) in mononuclear cells, thereby allowing mobilization into the circulation [46]. Likewise, endothelial-derived EVs are increased after acute MI and trigger mobilization and activation of monocytes from the spleen [47]. Thus, EVs derived from cardiomyocytes and endothelial cells recruit peripheral mononuclear cells and alter their phenotype. Furthermore, the number of EVs closely correlates with the extent of myocardial injury [47], suggesting a potential diagnostic and prognostic value of EVs in MI. Interestingly, EVs derived from endothelial cells overexpressing Krüppel-like factor 2 (KLF2) inhibit the recruitment of Ly6C^high^ monocytes and attenuate ischemia–reperfusion injury following MI [48]. EVs released by cardiac stromal-progenitor cells in mice, rats, and humans also have immunomodulatory effects through macrophage polarization in models of MI, highlighting their therapeutic potential [49,50,51]. 

Importantly, EVs derived from non-cardiomyocytes can modulate the inflammatory and fibrotic responses after MI. In innate immunity, macrophages release various non-coding RNA species that target cardiac fibroblasts. For example, EVs containing the circular RNA circUbe3a exacerbate myocardial fibrosis by altering cardiac fibroblast proliferation, migration, and phenotype (Figure 3) [52]. Activated macrophages also transfer miR-155-laden EVs to cardiac fibroblasts, suppressing their proliferation, promoting inflammation, and ultimately leading to cardiac rupture [53]. Additionally, the transfer of miR-155-containing EVs from macrophages to endothelial cells exerts an anti-angiogenic effect, further exacerbating ischemic injury [54]. On the other hand, EVs derived from cells of innate immunity can have salutary effects in acute MI. Following MI, EVs from dendritic cells mediate the activation of CD4^+^ T cells, and thereby improve cardiac function after MI [55]. Therapeutic modulation of T cells in MI has recently gained considerable interest [56,57,58]. Using regulatory T cells (Tregs) and their EVs for such modulation is a promising strategy. Indeed, EVs released by cardiac stromal-progenitor cells have been shown to potentiate Tregs by increasing their proliferation and IL-10 production, with downstream cardioprotective effects in models of inflammation [59]. Thus, EVs have diagnostic, prognostic, and therapeutic potential in models of MI and ischemic heart disease. Further translational studies are needed to potentiate their clinical value. So far, we have demonstrated that EVs play a significant role in the pathophysiology of atherosclerosis, which, untreated, can lead to ischemic heart disease. Ischemic heart disease, in turn, can progress to ischemic cardiomyopathy and heart failure. 

### 2.3. The Role of EVs and Inflammation in Heart Failure

Heart failure (HF) is a clinical syndrome caused by structural and/or functional abnormalities of the heart [60]. With a global prevalence of 64.3 million in 2017, HF is a major public health concern [61,62]. In the United States, the prevalence of HF is projected to increase to >8 million by 2030 [63]. The proportion of HF patients with reduced ejection fraction (<40%) ranges from 36% to 81%, depending on the region and registry used [62]. The proportion of HF patients with preserved ejection fraction (≥50%) ranges from 16% to 52%, but the prevalence appears to be increasing [64]. Both the left and the right heart can fail [65], and the causes can be broadly classified as either ischemic or non-ischemic. The mechanisms behind heart failure are varied and depend on the underlying cause. Nevertheless, EVs have been broadly implicated in several aspects of HF pathophysiology, including those that involve chronic inflammation.

Circulating EVs and their cargo miRNAs have been identified as potential biomarkers and drivers of HF [66,67,68,69]. A subset of these EVs originate from the inflammatory processes that accompany the initial insult and may contribute to the progression of HF. In animal models of MI, there is an increase in circulating EVs, carrying IL-1α, IL-1β, and RANTES (Figure 4). The reduction in these pro-inflammatory EVs (originating from M1 macrophages) results in improved cardiac function [70]. Notably, the interaction between cardiomyocytes and macrophages is bidirectional. For instance, in animal models of cardiac hypertrophy, EVs from hypertrophic cardiomyocytes activate macrophages by transferring miR-155 (Figure 4) [71]. This activation triggers the release of pro-inflammatory cytokines, IL-6 and IL-8, from macrophages. 

Because inflammation and adverse cardiac remodeling are interrelated [72], the role of fibroblast-derived EVs is equally important. In vitro, fibroblasts stimulated with TNF-α release EVs containing miR-27a, miR-28-3p, and miR-34a, which dysregulate the nuclear factor erythroid 2–related factor 2 (Nrf2) pathway that ordinarily prevents oxidative injury and protects against adverse cardiac remodeling and dysfunction (Figure 4) [73]. Cardiac fibroblasts also secrete EVs containing miR-21* and miR-27a*, which induce cardiac hypertrophy [74,75,76,77]. Cardiomyocytes, in turn, secrete miR-217-containing EVs that enhance fibroblast proliferation and drive cardiac hypertrophy [67]. In the context of ischemia, long non-coding RNAs in cardiomyocyte-derived EVs interact with fibroblasts and increase the expression of profibrotic genes [78]. Therefore, the underlying mechanisms in heart failure due to ischemic and hypertrophic etiologies are an active interplay between macrophages, cardiomyocytes, and fibroblasts. 

### 2.4. The Role of EVs and Inflammation in Aneurysmal and Valvular Pathology of the Aorta

The prevalence of thoracic aortic aneurysms (TAA) (>5 cm), incidentally noted on CT imaging performed for screening, ranges from 0.2% to 0.3% [63]. The overall prevalence of abdominal aortic aneurysms (AAA) ranges from 4% to 8%, but the prevalence of AAA ≥ 5.5 cm ranges from 0.4% to 0.6% [79,80,81,82,83]. Aortic aneurysms in younger individuals are typically due to hereditary or familial syndromes (e.g., Marfan syndrome, Loeys-Dietz syndrome, bicuspid aortic valve disease), while in older individuals the aneurysms are degenerative and share many risk factors with atherosclerosis [84]. Despite differences in the pathophysiology between TAAs and AAAs, EVs are clinically relevant in both as diagnostic and prognostic biomarkers and provide important insight into the underlying mechanisms.

Proteomic analysis of human plasma identified differential profiles in patients with aortic aneurysms, including proteins that drive inflammatory processes [85]. Platelets are activated in both TAAs and AAAs [86,87], and EVs derived from these platelets contribute to the pathogenesis of aortic aneurysms. For example, increased ficolin-3 levels are noted in platelet-derived EVs in patients with aortic aneurysms, compared with healthy controls, and are associated with aneurysm progression [88]. Ficolin-3 is one of the recognition molecules within the lectin pathway that activates the complement system, which, in turn, mobilizes innate immunity [89]. EVs from leukocytes are also identified in aortic aneurysms. Macrophage-derived EVs are found in murine and human adventitia from aneurysmal tissue. In vitro, these EVs induce the expression of matrix metalloproteinase-2 within vascular smooth muscle cells via the JNK and p38 pathways [90]. Neutrophil-derived EVs may also play a role in aneurysm progression. EVs bearing neutrophil markers are identified within the intraluminal thrombus of aortic aneurysms. These EVs carry active proteases ADAM10 and ADAM17, which contribute to aortic wall degradation [91]. Finally, the interaction between adaptive and innate immunities must be considered in aneurysm development. Activated T lymphocytes within vascular lesions are noted to have increased pyruvate kinase muscle isozyme 2 (PKM2) expression [92]. Importantly, these lymphocytes release EVs that are rich in polyunsaturated-fatty-acid-containing phospholipids and are taken up by macrophages. The activated lymphocyte-derived EVs then promote macrophage migration, lipid peroxidation, and iron accumulation, which drive aortic aneurysm progression. Corollary findings are noted in patients with AAAs, highlighting the translational value of these findings [92]. Thus, inflammation plays a prominent role in aortic aneurysm development, especially AAAs. EV interactions with non-immune mediators, however, are equally important in understanding the pathophysiology of aortic aneurysms. Specifically, vascular smooth muscle cell biology is central; the reader is referred elsewhere for focused reviews on this topic [84,93].

The prevalence of valvular disease depends on severity. Mild valvular disease is present in 51% of individuals ≥ 65 years of age, and moderate or severe disease is present in 6.4% [63]. The prevalence also varies based on specific valvular pathology, with aortic valve stenosis (AS) noted in 4.3% of individuals ≥ 70 years of age [63]. Chronic inflammation, infiltration of mononuclear cells, and activated EVs are all implicated in AS [94,95,96]. In a study of 22 patients with severe AS, leukocyte-derived EVs, platelet-derived EVs, and endothelial-derived EVs were all elevated, compared with healthy controls. Furthermore, these elevations were associated with monocyte activation [97]. Indeed, calcified aortic valves in humans have increased infiltration of macrophages that colocalize with metalloproteinases, which are involved in AS pathogenesis [98]. Macrophages may also contribute to valvular microcalcification through mechanisms discussed above in atherosclerotic disease [39,40,99]. Non-leukocyte EVs (e.g., endothelial cells) also play a critical role in aortic valve pathology, and the reader is referred to previous reviews for more details regarding interactions between EVs, valve endothelial cells, and valve interstitial cells [100,101]. The role of proteoglycans as a nidus of calcium accumulation in matrix vesicles is one such example. In conclusion, both aneurysmal and valvular pathologies of the aorta involve complex interactions between leukocytes, non-inflammatory cells, and EVs derived from both. Further investigation of these interactions and their mechanisms will yield clinically relevant data.

## 3. Conclusions

Inflammation is involved in nearly all cardiovascular conditions, contributing to the pathogenesis at multiple stages of the disease. An increasing body of literature implicates EVs derived from leukocytes and non-leukocytes in this pathophysiology. In this review, we summarize the mechanistic role of EVs at the intersection of inflammatory processes and cardiovascular disease. In atherosclerosis, EVs are involved in the recruitment of monocytes, polarization of macrophages, and microcalcification, which together contribute to plaque formation, maturation, and destabilization. Following the natural progression of untreated atherosclerosis, we then describe the role of EVs in ischemic heart disease, where EVs interact locally in the heart and peripherally in the bone marrow and spleen to recruit and modulate mononuclear cells. Following the progression of untreated cardiac ischemia, we discuss heart failure, where EVs derived from inflammatory cells, cardiomyocytes, and fibroblasts shape cardiac remodeling. Finally, we briefly discuss the role of EVs in aortic aneurysmal and valvular pathology, where EVs are implicated in inflammatory processes that influence proteinase activity and calcification. The importance of EVs, however, extends beyond their role in inflammatory processes. Throughout the text, we refer the reader to other sources that discuss EV involvement in non-inflammatory mechanisms of cardiovascular disease. It is important to note that much of the EV data summarized in this review (and others) are derived from preclinical models, often from isolated in vitro assays and small animal models. Furthermore, much of the data were generated before the minimal information for studies of extracellular vesicles guidelines were established by the ISEV. Given various isolation methods (e.g., differential centrifugation, density gradient centrifugation, size exclusion chromatography, ultrafiltration, immunoprecipitation, immunocapture, precipitation, and others), variable yields and purities, lack of universally accepted EV-specific markers, and diverse bioactive cargo, further rigorous studies are needed to strengthen the translational potential of the findings presented in this review. Nevertheless, EVs are promising biomarkers for diagnostic and prognostic applications. 

Finally, our discussion centered primarily on the diagnostic and prognostic value of EVs, with little mention of therapeutics. Understanding the mechanisms and contribution of EVs to cardiovascular pathology can in principle identify novel therapeutic targets. For instance, biogenesis and uptake of EVs is facilitated by diverse but targetable pathways (e.g., heparan sulfate proteoglycan pathway), which can be modulated for therapeutic purposes. Furthermore, EVs derived from various sources have emerged as promising cell-free therapeutic agents [102,103,104]. As reviewed previously [104], the potential advantages of EVs over cell therapy include EV stability, relative non-immunogenicity, and capacity for pre- and post-isolation modification. Therefore, EVs and their cargo will play an important role in the development of cell-free cardiovascular therapeutics.

## Figures and Tables

**Figure 1 cells-11-02229-f001:**
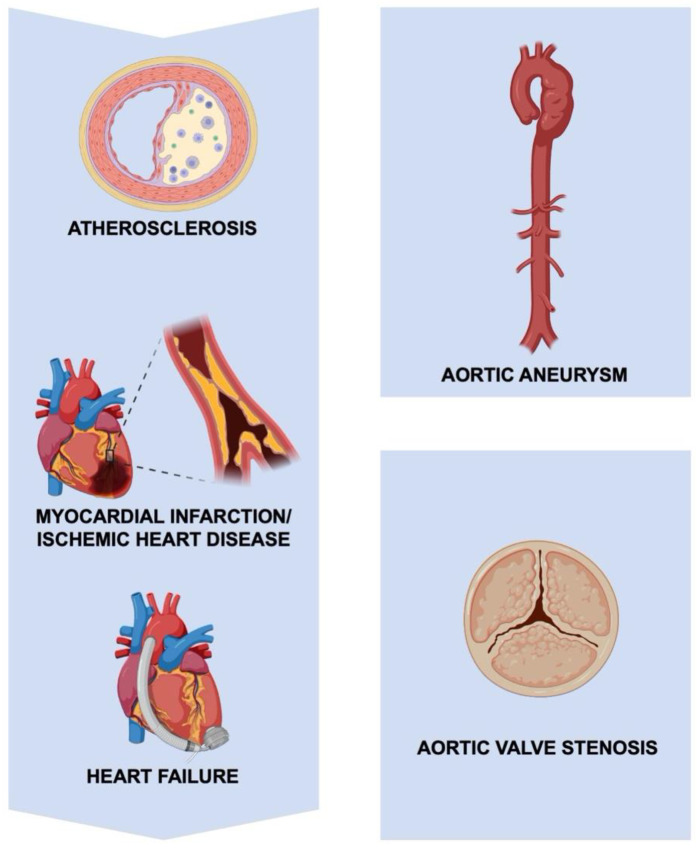
**Cardiovascular diseases.** Cardiovascular diseases, including the progression of atherosclerosis to ischemic heart disease and heart failure, aortopathy, and valvular pathology.

**Figure 2 cells-11-02229-f002:**
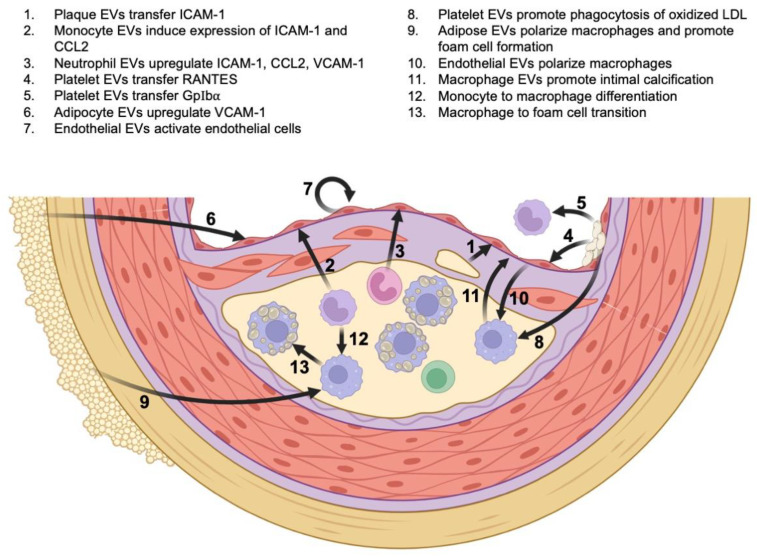
**The role of EVs and inflammation in atherosclerosis.** Mechanistic links between EVs, inflammatory mediators, and atherosclerotic plaque progression.

**Figure 3 cells-11-02229-f003:**
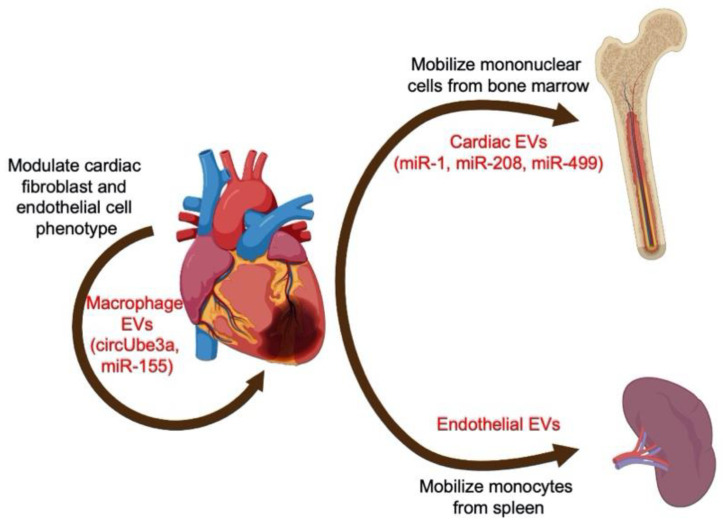
**The role of EVs and inflammation in myocardial infarction.** Crosstalk between cardiomyocytes and local and peripheral inflammatory cells.

**Figure 4 cells-11-02229-f004:**
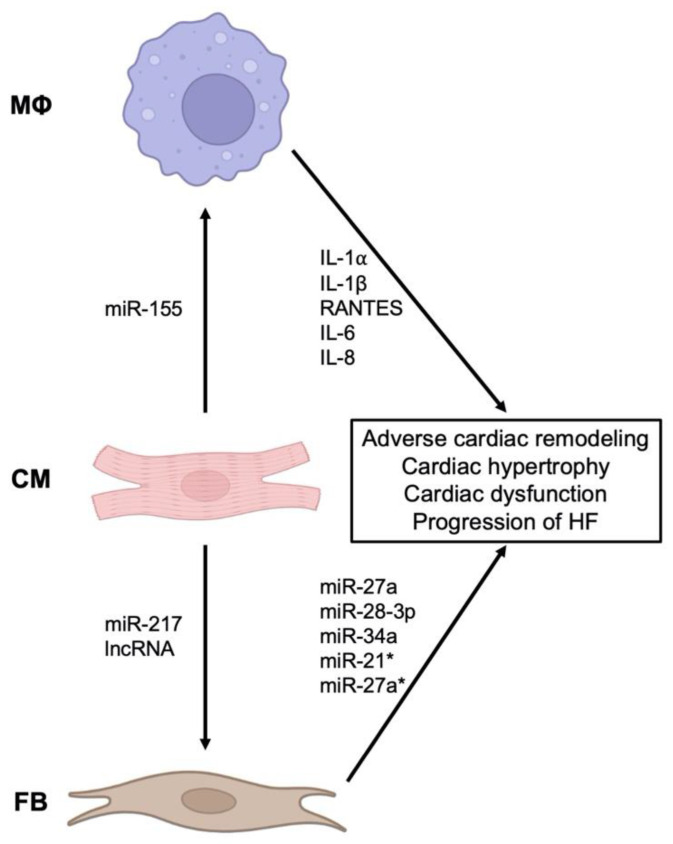
**The role of EVs and inflammation in heart failure.** Interaction between cardiomyocytes (CM), macrophages (Mϕ), and fibroblasts (FB) through EVs in the pathogenesis of heart failure.

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
