# Peer review of "Extracellular Vesicles, Inflammation, and Cardiovascular Disease"

_cells, 2022, doi:10.3390/cells11142229_

Round 1

Reviewer 1 Report

The manuscript from Akhmerov and Parimon is an interesting analysis of the literature of a subject of growing importance in cardiovascular science. Nonetheless, extensive reorganising and perhaps refocussing is required to be suitable for publication and to truly benefit the readers of Cells. Here are my suggestions to improve the manuscript:

  1)     The division in diseases ((i.e. atherosclerosis, MI, HF, TAA, etc) is not sufficient. The authors should consider titled subsections based on target cells or originating cells.

  2)     The description of the different studies important in the field is interesting, but: a) it is currently too concise to help the reader understanding, b) more efforts should be made to cover all of the literature (currently it is not clear how the authors focus on some studies and omitted others).

  3)     The previous point brings me to this point: the field is already quite diverse and developed, therefore the amount of material to cover seems too much for one literature review (leading to omission of some studies and lack of details in the exposition). The authors should consider focussing on only one disease and generate a literature review specific to the chosen disease (e.g. “involvement of MVs in the progression of MI”, etc.).     

  4)     Figures and tables will be required to facilitate comprehension and the synthesis of the text. Without figures and tables, the text remains a fairly cryptic list of concisely quoted studies. Its fruition would be problematic.  

  5)     The conclusion is very general. It would be great to focus on clinical opportunities and the hurdles hindering their exploitation.

  6)     A section on technical challenges in this field explaining why MV-based clinical applications remain largely unexplored would be good.

Author Response

1)     The division in diseases ((i.e. atherosclerosis, MI, HF, TAA, etc) is not sufficient. The authors should consider titled subsections based on target cells or originating cells.

We thank the reviewer for this suggestion.  However, dividing the manuscript based on cell type will fragment the manuscript into a less cohesive narrative.  The divisions would include macrophages, monocytes, platelets, adipocytes, endothelial cells, cardiomyocytes, neutrophils, T cells, discussed in a context of disparate pathologies.  In our structure, we were hoping to provide a linear natural-progression-of-disease narrative (i.e., atherosclerosis leads to MI, which leads to heart failure, etc), which is heuristically useful not only to the basic scientist but also to the physician scientist.  As such, we believe this structure will have broader impact.  We have included a central figure to highlight this organization (Figure 1).

  2)     The description of the different studies important in the field is interesting, but: a) it is currently too concise to help the reader understanding, b) more efforts should be made to cover all of the literature (currently it is not clear how the authors focus on some studies and omitted others).

      We thank the reviewer for this observation.  However, given the deliberately focused and narrow scope of our topic, this review is comprehensive for the pathologies described.  Importantly, this is the only review to our knowledge that directly focuses on EVs + inflammation + CV disease, in combination.  All prior reviews have focused on one of these in combination with one other (e.g., EVs + inflammation but not CV disease; EVs + CV disease but not inflammation), so our review has a narrow focus that explicitly incorporates all three concepts.  This is explained in the last paragraph of the Introduction, page 1, lines 37-41.  As such, the literature we used includes only those references that directly support the interaction of all three elements (EVs + inflammation + CV disease).  We omitted conjectures and conceptual extrapolations that are not directly supported by data.  We invite the reviewer to suggest any important studies that were omitted/missed that explicitly address all three concepts (EVs + inflammation + CV disease), and we would be glad to incorporate these studies into our revisions.

  3)     The previous point brings me to this point: the field is already quite diverse and developed, therefore the amount of material to cover seems too much for one literature review (leading to omission of some studies and lack of details in the exposition). The authors should consider focussing on only one disease and generate a literature review specific to the chosen disease (e.g. “involvement of MVs in the progression of MI”, etc.).    

      Please response to comment 2. 

  4)     Figures and tables will be required to facilitate comprehension and the synthesis of the text. Without figures and tables, the text remains a fairly cryptic list of concisely quoted studies. Its fruition would be problematic.  

      Schematic figures for each section have been added to facilitate comprehension.  We hope these visual aids make the interactions between inflammatory mediators, EVs/EV cargo, and cardiac pathology more clear.

  5)     The conclusion is very general. It would be great to focus on clinical opportunities and the hurdles hindering their exploitation.

      We have modified our conclusion, which addressed clinical opportunities and hurdles.  See lines 265-282 in conclusion.

  6)     A section on technical challenges in this field explaining why MV-based clinical applications remain largely unexplored would be good.

      See comment for #5.

Reviewer 2 Report

This is a very interesting review on the EVs during cardiovascular diseases and inflammation. The review is well-structured and English language is correct.

However, in this review there is an important lack of figures that could depict the various topics, ex. the role of EVs from various cell types during atherogenesis.

Moreover, authors do not include the role of proteoglycans and glycosaminoglycans, such as heparan sulfate, in cardiovalscular diseases and EVs.

Lastly, it would be helpful to include the whole name of the abbreviations that they use at their first citation, ex. ICAM-1, NF-kappaB etc

Author Response

This is a very interesting review on the EVs during cardiovascular diseases and inflammation. The review is well-structured and English language is correct.

However, in this review there is an important lack of figures that could depict the various topics, ex. the role of EVs from various cell types during atherogenesis.

Thank you for this excellent suggestion.  We have incorporated figures for the major topics (Figure 1, Figure 2, Figure 3, Figure 4).

Moreover, authors do not include the role of proteoglycans and glycosaminoglycans, such as heparan sulfate, in cardiovalscular diseases and EVs.

Thank you for this suggestion.  We have mentioned the importance of proteoglycan in the valvular pathology section, by mentioning “the role of proteoglycans as a nidus of calcium accumulation in matrix vesicles.” (Lines 244-245)  We also mentioned proteoglycan pathways contributing to the biogenesis and uptake of EVs, thereby providing potential therapeutic targets (Lines 278-282).

Lastly, it would be helpful to include the whole name of the abbreviations that they use at their first citation, ex. ICAM-1, NF-kappaB etc

      Thank you for this suggestion.  We have included the full names at first mention throughout the text (line 62-63, line 65-66, line 68, line 71, line 186-187)

Round 2

Reviewer 2 Report

The authors have modified the text according to the comments of the reviewers. The manuscript is now presented better. However, there are some minor changes to perform.

Titles and subtitles of section 2 should contain the keywords EVs and/or inflammation, as described in the last paragraph of the Introduction, otherwise it seems that the review describes generally the diseases. For example, Inflammation and EVs role in different Cardiovascular disorders. 

There is still no description of the role of proteoglycans and glycosaminoglycans (such as heparan sulfate) in cardiovascular diseases. The sentence inserted is very general; a paragraph with a description of which proteoglycans/glycosaminoglycans are impicated in these diseases is missing.

Legends of the figures are missing.

In the figure 4, the abbreviations of the cells used are not described in the text (and legend). Is FB fibroblast? And the others, what are they? Please, cite at least in the legend.

Author Response

Thank you for these suggestions.  We have modified titles and subtitles according to the reviewer's recommendation.  An additional paragraph regarding proteoglycans was added in the discussion of atherosclerosis pathogenesis (lines 102-112).  Figure legends have been uploaded.